# Weeds Enhance Pollinator Diversity and Fruit Yield in Mango

**DOI:** 10.3390/insects12121114

**Published:** 2021-12-13

**Authors:** Blaire M. Kleiman, Suzanne Koptur, Krishnaswamy Jayachandran

**Affiliations:** 1Department of Earth and Environment, Agroecology Program, Institute of Environment, International Center for Tropical Botany, Florida International University, 11200 SW 8th St, Miami, FL 33199, USA; bkleiman@fiu.edu; 2Department of Biology, Plant Ecology Lab, Institute of Environment, International Center for Tropical Botany, Florida International University, 11200 SW 8th St, Miami, FL 33199, USA; 3Department of Earth and Environment, Agroecology Program, Institute of Environment, Florida International University, 11200 SW 8th St, Miami, FL 33199, USA; jayachan@fiu.edu

**Keywords:** weeds, insects, mango, pollinators

## Abstract

**Simple Summary:**

There is an urgent pollinator decline crisis across the globe, with fewer pollinators and yet increasing agricultural reliance on them to produce food and fiber crops for growing populations. Habitat loss and chemical eradication of unwanted plants has limited the floral resources for pollinators, and in farms with only one crop, there are limited resources solely during the flowering season. Weeds, or unwanted vegetation, are often the only remaining floral resource for pollinators, yet they are compulsively removed using chemicals. This article examines how weedy floral resources affect pollinators in a mango farm, *Mangifera indica*, a pollinator-dependent crop in South Florida, and how fruit yield is affected by either leaving weeds or removing them.

**Abstract:**

Agriculture is dependent on insect pollination, yet in areas of intensive production agriculture, there is often a decline in plant and insect diversity. As native habitats and plants are replaced, often only the weeds or unwanted vegetation persist. This study compared insect diversity on mango, *Mangifera indica*, a tropical fruit tree dependent on insect pollination, when weeds were present in cultivation versus when they were removed mechanically. The pollinating insects on both weeds and mango trees were examined as well as fruit set and yield in both the weed-free and weedy treatment in South Florida. There were significantly more pollinators and key pollinator families on the weedy mango trees, as well as significantly greater fruit yield in the weedy treatment compared to the weed-free treatment. Utilizing weeds, especially native species, as insectary plants can help ensure sufficient pollination of mango and increase biodiversity across crop monocropping systems.

## 1. Introduction

Cultivated crops are often subject to insect–plant interactions for high yield. There has been a growing interest in environmentally and ecologically sound agriculture using beneficial insects rather than pesticides to produce food and fiber without harmful chemicals in produce and the environment [1,2]. Ecological intensification is the use of biological regulation to manage agroecosystems at various scales [3]. Natural ecosystems can inspire cropping system designs [4], and these approaches may have greatest impact in high-input farming systems [5,6]. Ecological replacement, substituting biodiversity for synthetic inputs, can enhance ecosystem services with similar crop output [6]. The presence of non-crop plants in planted floral strips may be useful in this approach, as a recent meta-analysis has shown [7]. Weeds may also provide resources that attract and maintain populations of beneficial insects, such as pollinators. Weeds—wild plants growing where they are not wanted—are seen as detrimental to crop production in agriculture by pulling resources away from the crop. This lack of weeds diminishes beneficial insects through the loss of floral and prey resources [8]. The benefits of using insectary plants in farms is well known [9,10]; however, using weeds as such in tropical fruit production dependent on pollination is relatively unexplored. Previous work has shown increased success of beneficial insects in the presence of weeds, as these insects use nectar or pollen during their adult life stage to increase life span and fecundity [11]. Pollinator populations may be bolstered in the presence of weeds, and some have been shown to be dependent on them [12,13]. This study examined how leaving, rather than removing, weeds in a mango farm affected pollinators and fruit set of this popular tropical fruit cultivated in southern Florida.

Pollinators are especially important in crops that require pollination by insects [14,15], such as mango, which is known to benefit from the presence of a diversity of weeds. Many tropical crops may be most susceptible to pollination failure from habitat loss [16,17]. Almost 35% of crops depend on pollinators globally, with pollination of at least 63 crops vulnerable to negative effects of agricultural intensification, which may reduce the diversity and abundance of pollinators [18]. The global annual economic value of insect pollination is upward of USD $173 billion worldwide [19]. Pollination by bees and other animals increases the size, quality, and stability of yields for 70% of leading economically important crops around the world [16], including mango [20]. Because native species pollinate many of these crops effectively, conserving habitats for wild pollinators within agricultural landscapes can help promote pollination services for most of the world’s crops [21].

There is a pollinator crisis in areas of intensive human land use and landscape simplification, including farmlands [22]. Insects have shown marked population declines over the past 30 years, with the average decline of terrestrial insect abundance at about 9% per decade [23]. Decline in pollinators is intertwined with habitat simplification through the expansion of monoculture practices [24] and increased applications of pesticides and fertilizers [25,26]. Pollinator abundance increases with flower abundance, vegetation height, and floral diversity [27]. The conservation of plant diversity safeguards native pollinator diversity as well as overall biodiversity and ecosystem services [28]. Mass flowering plants can act as “pollinator hogs”, which can reduce the pollination success of adjacent co-flowering neighbors by drawing pollinators from these plants [29,30]. However, mass-flowering plants can also act as magnets, producing pollination “spillover effects” through increased pollinator movements to adjacent co-flowering taxa, potentially either increasing pollination [31] or impacting it through the transfer of mixed-species pollen [32]. The presence of flower diversity before and during crop flowering facilitates pollination of the hyperabundant crop flower resource [20,33]. For a pollinator-dependent crop, creation of flowering areas can be profitable, improving production within existing areas and reducing the need for agricultural expansion, contributing to the conservation of biodiversity within a region, and increasing the habitat and resources for insects within farms [34].

Weeds have the potential to increase biodiversity of native pollinators by providing alternative floral resources for beneficial insects and encouraging them to remain in an area between crop flowering events [14,35,36]. Pollinators can use weeds as alternative resources before, during, and after the bloom of a crop, and increase crop yields if given these resources [37]. Weeds are an essential pollen and nectar resource for insects because of their continuous flowering phenology and their high species richness, which contributes directly to the pollen diversity dietary needs of insects [28]. Pollinators are healthiest, with at least 15 flowering species providing a season-long food supply [38], and weeds can provide this floral diversity. Arable weeds have specific functional traits that make them tolerant to farmlands, such as soil disturbances and fertilization, making a large overlap in the weedy potential of non-weed species.

Effective insect pollination is essential for good fruit set and yield in mango (*Mangifera indica* L., Anacardiaceae) [39]. Mango flowers are unspecialized, enabling pollination by most insects that are critical for fruit yield [40]. Nectar production for the attraction of insects indicates entomophilous pollination, and mango does not show adaptations for wind pollination [41]. Managed pollinators are unsuitable [42] or insufficient when acting alone [20,43,44]; honeybees are generally not attracted to mango flowers [45,46], and hand pollination is not economically viable [47]. Pollination is highly dependent on a diverse assemblage of flying visitors, which is strongly negatively affected by distance to natural habitat [46]. Small patches of native flora, planted in nonproductive margins of large mango orchards, enhance the abundance and diversity of mango flower visitors in South Africa, ameliorating the negative effects of isolation from natural habitat and pesticide use [34]. These increases were associated with significantly higher mango production, including the Keitt variety. Two co-flowering species can compete with or facilitate each other for flower visitors, although some studies suggest that facilitation is more likely between plants with unequal flower abundance [29] or that attractive species may facilitate less attractive species, as is the case with mango [20,33].

The insect orders Diptera, Hymenoptera, Lepidoptera, and Coleoptera are the most common insect visitors to mango flowers and carry mango pollen [39,41]. Most pollinators of mango belong to order Hymenoptera, but Diptera (flies) have been suggested as the dominant pollinators in Israel [39]. Dipteran species are good pollinators of more than 550 species of flowering plants, and the family Calliphoridae (blowflies) are the suspected main pollinators of mango in India [48]. Additionally, *Chrysomya, Lucilia*, and *Musca* sp. (Diptera) were reported as mango pollinators because of their visiting frequency and abundance [48]. Flower flies (Syrphidae) are also good pollinators of mango [49], as they transport pollen for long distances and reproduce rapidly [50]. The introduction of three pollinators—the honeybee, the bumblebee (*Bombus terrestris*), and the housefly—resulted in much higher yield of mango vs. caged pollinator-free trees [39]. Common blow flies may also serve as effective mango pollinators and are less likely than honeybees to abandon the mango orchard for more attractive blooms, in Israel as well as in the United States [39].

The aim of this study was to examine if weeds can increase biodiversity of pollinators and ultimately benefit Mango (*Mangifera indica*) crop cultivation. We asked the following questions:How do the abundance and diversity of insect species (potential pollinators) differ on mango in the presence or absence of weeds?What is the impact of weeds on mango fruit yield?

If weeds benefit mango trees, we anticipate a higher abundance and diversity of pollinating insects on the mango trees where weeds have been left to grow (weedy treatment) in comparison to mango trees that have been cleared of all weeds (weed-free treatment). Perhaps this enhanced pollinator abundance will increase fruit production.

## 2. Materials and Methods

### 2.1. Site Description

The field experiment was installed at a conventional mango farm (20 acres), variety “Keitt”, within the major agricultural area of Homestead, Florida (25°29′42.9″ N 80°29′30″ W). Trees were evenly spaced in 20 × 20 feet intervals. Distance between rows was approximately 20 feet. Two treatments were applied to the trees: weedy vs. weed-free. Three sections of 10 trees were assigned per treatment, with buffer rows and trees surrounding the treatments (Figure 1). For the weedy treatment, unadulterated weed growth was allowed between the trees, and weed species identified. All were in the same area of the site as per maintenance requirements of the farmer. For the weed-free treatment, weeds were removed around the crop, using a mower and string trimmer, a tool for cutting grasses, small weeds, and vegetation. All weed specimens within the weedy treatment were vouchered and identified.

### 2.2. Farm Maintenance

The mango cultivation area of this farm is made up of 24 rows with 47 mature mango trees each and a mix of Tommy, Keitt, Kent, and Florida Red varieties. The weed treatment was placed in row 24 (Figure 1), with buffer trees separating the sections, due to restrictions in field maintenance of separating the treatment across multiple rows. The 3 weed-free sections were assigned to 10 trees each in rows 2, 4, and 6.

The management practices used at the study site are ecologically oriented and minimal for a large conventional farm compared to other mango farmers in the area, and no insecticides are used by the grower to allow balanced insect biological control of pests. Similarly, mango farmers in the United States are limited in what chemicals they can apply to trees compared to those in other countries; for example, the use of Topsin to treat mango malformation is outlawed in the US [51]. The major chemicals applied to the trees by the grower were fungicides to kill anthracnose, bacterial, and fungal pathogens. Before fruiting, micronutrients were added to all the trees as well as cow manure instead of synthetic fertilizers. Mowing occurred in the farm as needed (outside of the weedy treatment) with a sickle bar mower beneath the trees, rather than herbicidal eradication of weeds.

### 2.3. Field Data Collection

Observations and collections of insects were made from the mango trees in both treatments. Insects interacting with the crop *M. indica* were recorded and collected weekly for six months including the major 8-week blooming period of mango. Five-minute watches were made recording insect interactions, observing, and collecting specimens on each of the 30 mango trees in both treatments totaling 7500 min of observations across the 25 weeks. Timed focal point observations were conducted weekly before, during, and after the peak mango flowering season (from November until May), similar to methods outlined in Carvalheiro et al. 2012 [34].

The timing of observations per treatment per day was changed for each data collection day by alternating the order each tree was visited, to have a breadth of observations across the day. Close focusing binoculars were used to observe specimens in the upper canopy of the mango trees if needed. In each watch period per tree, all insects and flower visitors were recorded and collected for identification. Each insect specimen was collected if novel or if sight identification was not possible. Additionally, specimens were collected if they displayed notable behavior such as pollinating flowers using an insect aspirator, collection bag, or net. Specimens collected from the field were immediately placed into a plastic bag that were kept cool and stored in a freezer (0 °C) until further identification. Voucher specimens were pinned or stored in ethanol in vials and are maintained in the authors’ collection; they will eventually be deposited in the Florida State Collection of Arthropods. There were two flushes of inflorescences on the mango trees during this season, allowing insects to be collected from mango flowers from 12/05/2019–05/08/2020.

### 2.4. Statistical Analyses

Statistical analyses were performed with SPSS software. Descriptive information is presented to describe the variables of interest overall and by treatment. These include means, standard deviations, frequencies, and proportions. To compare means between treatments, a two-sample t test was used, and to compare proportion, chi-square test was used to see if the observed distribution of insects differed from the expected distribution. Statistical analyses of effect of treatment on insects compared between the weedy and no-weed treatment, as well as these effects on fruit yield, was performed using a general linear model by applying multivariate analysis of variance (MANOVA) with insect/fruit yield measures as the dependent variable (multiple Y’s) as a function of treatment while adjusting for tree age. Multivariate tests such as Pillai’s Trace, Wilk’s Lamba, Hotelling’s Trace, and Roy’s Largest Root examined overall model significance, followed by further analyses using simplified F-test comparison adjusting for multiplicity. All differences or associations were considered significant at the alpha level of 0.05 after the adjustment for multiplicity where appropriate.

### 2.5. Fruit Yield

Fruit counts per tree were performed visually when nearly mature fruit were set on the tree as well as counted when harvested by the grower. Visual fruit yield counts were performed after bloom and all fruit had been pollinated and set (number of unripe fruits per tree, circa 6 weeks after the end of flowering ceased, as in Carvalheiro et al., 2012 [34]. For each visual count, two different observers’ counts were averaged. A marker for the starting point of observation was placed on the ground, and an observer used a hand clicker counter and moved around the tree counting each individual mango fruit. Early fruit set is only an indication of pollination efficiency but may not be a good indicator of pollination quality and, hence, of final crop production and economic value [34]. Therefore, we also obtained information on final production as pounds of commercially suitable mangoes directly from the farmer. Fruit was harvested green for the Asian market early in the season in May and July. Harvest of mango is performed manually by pickers; therefore, a count was to be taken as each tree was picked.

## 3. Results

This was part of a larger study in which all insects associated with mango and weeds were collected. There were 3786 total records of flower visitors observed and/or collected during this study.

### 3.1. Flower Visitors of Mango

When comparing arthropod numbers on mango trees in both treatments, there was a significant effect of treatment (weeds) on the number of individual visitors to mango flowers (F = 31.109, df = 1, 57, *p* < 0.0001; Figure 2) determined by multivariate generalized linear model analysis, Table 1). There were significantly more insects feeding in mango flowers in the weedy treatment than the weed-free treatment (Table 1, F = 45.93, df = 1, 56, *p* < 0.0001).

There were significantly more Hymenoptera on weedy mango trees (Table 2, F = 80.07, df = 56, *p* < 0.001) (Figure 3).

Additionally, flies were abundant flower visitors, and there were (nearly significantly) more on weedy mango trees than weed-free ones (Table 2, F = 3.79, df = 1, 56, *p* = 0.057; Figure 4). There were significantly more thrips (Table 2, Thysanoptera F = 8.208, df = 1, 56, *p* = 0.06) on the weed-free mango trees than the weedy treatment trees. There was a significant difference in beetles (Coleoptera) F = 10.59, df = 1, 56, *p* = 0.002, with more feeding in the mango flowers with weeds present.

There was no significant difference in the order Lepidoptera (butterflies, skippers, and moths) (Table 2), potential mango pollinator insects, between weedy and non-weedy mango trees [41]. There was, however, a significant difference in the family Lycaenidae, the second largest family of butterflies, including the hairstreaks and blue butterflies, which were significantly greater in number on mango when weeds were present, T = 2.18, df = 48, *p* = 0.031 (Table 3).

### 3.2. Families

Insect family observations were aggregated by individual mango trees from both treatments; the aggregates were compared between the two groups using a t test. Of the total 126 insect families observed on the weed-free and weedy mango trees, 10 families differed significantly on the mango trees when weeds were present or not (Table 3).

The representation of many mango pollinating insect families was significantly greater on mango trees with weeds present: Apidae (bees) T = 4.08, df = 55, *p* =< 0.001, Calliphoridae (blowflies, Figure 5a) T = 2.03, df = 53, *p* = 0.048, Muscidae (housefly) T = 3.5, DF = 57, *p* = 0.001, and Syrphidae (hoverfly) T = 3.34, df = 44. *p* = 0.002. Calliphoridae is especially an abundant and effective mango pollinator who preferred to visit the mango flowers in the weedy treatment. Flower visiting insects such as Vespidae (wasps), T = 2.2, df = 18, *p* = 0.041 and Chalcididae (parasitoid wasps), T = 2.61, df = 29, *p* = 0.003, were also present in greater numbers in the weedy treatment.

Other insect families had significantly greater numbers on the weed-free mango trees, such as Chironomidae (non-biting midges) T = −2.5, df = 58, *p* = 0.016 and Drosophilidae (fruit/vinegar/pomace fly) T = −2.68, df = 33, *p* = 0.011. While these insects were observed visiting flowers in small numbers, they do not serve as key mango pollinators. There was no significant difference in the number of ants (Formicidae: Hymenoptera, Figure 5b) between weedy or weed-free mango trees (F = 3.89, df = 1, 22, *p* = 0.19).

### 3.3. Fruit Yield

There were significantly greater counted (F = 181.317, df = 1, 57, *p* < 0.0001) and harvested (F = 89.344, df = 1,57, *p* < 0.0001) fruit from trees in the weedy treatment vs. those that were weed-free (Table 1). Age (D. Lyons, pers. comm.) of mango trees was considered as a covariate but was not significant. Fruit count visual surveys by the two observers were averaged per tree, and the harvested count (Figure 6) from the two harvest days added together per tree. There was substantially more fruit when weeds were present vs. the weed-free treatment. Both visual counts of fruit set and harvested counts were significantly higher in the weedy treatment (Figure 7). The mean number of harvested fruit per tree in the weedy treatment was 179 ± 65 versus 38 ± 15 in the weed-free treatment. The mean number of counted fruit in the weedy treatment was 236 ± 100 vs. 48 ± 38 in the weed-free treatment.

### 3.4. Weed Species

Weed species for the weedy treatment were vouchered and identified. Seventy-five different species from 27 families were identified (Table 4). Of these species, 34 were native and 38 non-natives, of which six are classified as Category I invasive species and five are Category II. Invasive exotic plants are termed Cat I invasive (Cat I) when they are altering native plant communities by displacing native species, changing community structures or ecological functions, or hybridizing with natives with documented ecological damage caused [52]. Cat II invasive exotics have increased in abundance or frequency but have not yet altered Florida plant communities to the extent shown by Cat I species. These species may become Cat I if ecological damage is demonstrated. Data from Florida Exotic Pest Plant Council (FLEPPC) and Atlas of Florida Plant Institute for Systematic Botany [52].

## 4. Discussion

The overall results indicate the successful implementation of weeds as insectary plants to increase pollinators and fruit yield. This could be from the added floral diversity provided by weeds, enticing insects to visit nearby less attractive co-flowering taxa. The added diversity of 75 different weed species (many of which are native flowering herbaceous plants) increased the floral diversity for beneficial insects without negatively competing with the crop. Mango is one of the most widely cultivated tropical fruits worldwide and one of several drought-tolerant plants with minimal nutrition supplementation needs [53]. Its deep taproot and added fruit yield with maturity of the tree allow reduced negative impacts from intercropped herbaceous vegetation, and reliance on pollination means net gains from added floral diversity of weeds and pollinating insects.

Hymenoptera preference of weedy mango flowers indicates that although bees (Apidae) tend to not be attracted to mango [46], if attracted to forage within mango fields, their activity can spill over and contribute to mango pollination. Added habitat diversity, nesting materials, and floral resources with weeds can create favorable habitats for hymenopterans. Diptera abundance and preference of mango flowers, as well as pollen carrying capacity, make them valuable pollinators of mango. Adding resources and diversity through weeds can be an effective strategy to bolster their populations, and in turn fruit yield. Flower strips have been shown to be effective in increasing pollination of crops but not hedgerows [7]. An important difference between the two may be the distance from the crop plants, and weeds below the mango trees can attract beneficial insects in close proximity.

Thrips are a threat to the production of mango, as mango flower thrips feed on petals, anthers, pollen, and floral nectaries, resulting in discoloration, malformation, weakening of the inflorescence, and reduction of fruit set [54,55]. Thrips also can cause “bronzing” of the fruit surface and can make fruit unsuitable for fresh marketing [56]. Flower thrips have a broad range of hosts, including weeds that provide refuge between mango flowering seasons. Thrips may also act as pollinators, however, and are considered as “pure pollinators” because unlike bees and butterflies, they carry pollen grains of only one plant species [57]. There is evidence of pollen carried by both adult and immature thrips in mango inflorescences and possible synchrony in the population dynamics of these species with mango flowering, and there is possible pollination potential of these species with *Frankliniella gardeniae* reported as a mango visitor [58].

Previous research states that weeds act as a reservoir of thrips, enhancing populations in farms [55]. However, our results indicate the contrary: there are fewer thrips on the weedy mango trees, potentially indicating that the presence of weeds can act as a trap plant, pulling thrips away from the mango trees and inflorescences when they bloom (Table 2). This interaction could also account for the significantly higher fruit yield in the weed treatment, as there were less flower thrips causing damage and reducing fruit set. Further research on mango-thrip synchronicities and weed-mediated interactions should be performed to elucidate their effects on crop production.

There were significantly more beetles (Coleoptera) feeding in the mango flowers with weeds present. Many beetles are herbivores but may also visit flower parts and transport pollen. Beetles are a dominant group of potential visitors and pollinators of more open, unspecialized flowers, such as mango flowers [40]. Cantharidae beetles have been reported to be active on mango blooms in both Israel and Costa Rica [59]. The beetles most frequently recorded in mango flowers in this study were the flower beetles (Scarabidae, *Cetoniinae*) and ladybird beetles (Coccinellidae). The mango flower beetle can be a pest of mango and avocado in South Florida, where large numbers of the beetles destroy the flowers and reduce the number of fruits produced [60]. Their numbers were small (only observed eleven times), and they were observed feeding in mango flowers only for a short period during March. Further investigation into beetles as potential mango pollinators, and their association with weedy flowers, is warranted.

Vespidae (wasps) and Chalcididae (parasitoid wasps) were present in greater numbers in the weedy treatment. These insects have high energy needs and feed in flowers for key resources such as nectar and pollen. Wasp pollination has been observed in Vespidae that act as pollen vectors and both generalist and specialist pollinators, even surpassing bees as pollinators in some cases [61]. Predatory and parasitoid wasps are valuable natural enemies that control insect pest populations and are especially important in agricultural systems. The increased presence of wasps in the weedy treatment, where more floral and prey resources were available, adds both types of ecosystem services to mango cultivation.

There was no significant difference in the number of ants (Formicidae: Hymenoptera) between weedy or weed-free mango trees. Ants have been found to be important in pollination of mango [45,62] and have been found to contribute to 50% of the early fruit set and are not influenced by distance to natural habitat [34]. Ants, however, are not mobile and are more likely to contribute to self rather than to cross-pollination, potentially leading to fruit abortion. We observed no difference in ants on mango trees or in mango flowers between treatments, suggesting that they are not affected by added floral and alternative resources provided by weeds.

Additionally, elimination of herbicide applications in the weedy treatment can reduce a farmer’s production costs, chemical use in the adjacent suburban neighborhood, and potential runoff and negative effects of chemicals on surrounding ecosystems. It also leads to a substantial gain in fruit yield in the weed treatment. Each Keitt fruit weighs on average 1.5–2.0 lbs (D. Lyons, pers. comm.). The average current price of a pound of Keitt mango across a spread of outlets, after deducting all marketing and packaging costs, is USD 0.143/lb [34]. As tree density was 30 trees in the weed treatment and implementation of weeds as insectary plants has no costs, it led to a gain of USD 908–1210 for all 30 trees, notwithstanding added gains from elimination of herbicidal applications. As farmers sell their product per pound and not per mango, weight is a good indicator of economic profit. Because production cost is mainly determined per hectare and is not influenced by volume of crop to be harvested, all increases in volume will have a positive impact on the economics of the crop [34].

However, given the constraints and application of the weed treatment in this experiment within one row of mangos and not spread across the farm, potential confounding abiotic factors such as sunlight/adjacent landscapes could have also impacted fruit yield. Further studies should clarify how field and surrounding weedy vegetation affects insect biodiversity and mango fruit yield, as well as soil abiotic and biotic functions and nutrition. Using other sources of insectary plants, such as sweet alyssum [9], or native herbs such as the Bahama Senna [63] could be an alternative for farmers to garner beneficial insects in South Florida. Additionally, farms in this area are surrounded by an increasingly urbanized landscape, and the relationship among urbanization, pollinator resource distribution, pollinator abundance, and pollination service provision are uncertain [64]. Further, pollen supplementation experiments comparing cross- and self-pollination would help clarify the importance of cross-pollination in mango and the role of insect movement in fruit yield.

Anecdotally, we also observed more small mammals and snakes in the weed treatment compared to the rest of the farm, as well as flies feeding on carrion (the bodies of deceased rodents). As pollinators require distinct and diverse resources throughout their life cycle, including larval habitat and adult food resources, the presence of rat carrion has been found to facilitate pollination services to plants by bolstering the pollinator community across an urban gradient. Rat carrion has been found to increase pollinator abundance more than two-fold, and plants received 11.2% greater pollination service across landscapes and higher viable seed set, especially in densely urban landscapes [64]. Mutualistic species with complex life histories can provide conduits between various levels of ecosystem processes. As blowflies/carrion flies (Calliphoridae) are one of the best pollinators of mango, weedy habitats can potentially serve dual purposes in supplying adult needs for floral resources as well as their larval needs for carrion by supporting small animal populations. Blow flies can be used as effective mango pollinators along with other flies and bees and, at the farm level, are pollinators that are cheap and easy to rear [65]

## 5. Conclusions

In summary, we found that weeds increase the diversity of pollinating insects on mango flowers, and mangos with weeds growing below produced more fruit than those without weeds. Weeds can provide ground cover that can support insect natural enemies [66,67], pollinators [35,68,69], birds [70], and increase biodiversity at the field and landscape level [71,72,73]. Conserving the biodiversity of plants and insects needs more focus now than ever, with increasing threats to farms, especially monocultures, in the face of climate change [74]. More biodiverse farms, hosting varied plants and insects, are more resilient and less vulnerable to stressors [74]. Using naturally occurring weeds, which have the potential to act as insectary plants when growing companion plants is not possible (and increasingly more difficult in changing climates), can play an important role in adding to the variety of diets for beneficial insects, aiding in the conservation of plants, and their plant-pollinator networks. Plant-feeding insects are an important cause of crop yield losses worldwide [75,76,77], and these losses are predicted to increase in response to climate change [78,79]. Weeds, therefore, can provide resources to maintain balanced insect–ecological dynamics, and allow for less chemical use of pesticides and herbicides, and more environmentally sound agriculture in South Florida and other parts of the world.

Overall, the use of weeds in increasing beneficial insects has shown promise [12,80,81]. However, a caveat to their use in agriculture is how to handle invasive weedy species. Selectively removing the noxious FLEPPC Category I invasive plants is recommended to slow their spread in South Florida (a major problem) and allow native flowering weeds to take over [52]. While increasing plant diversity and resources to increase pollination may not always prove a success, studies on specific crops, regions, and weed species can allow us to learn how these variables affect plant–insect ecological dynamics. Furthermore, with increased pollination and crop yield, economic valuation can allow us to gain insight on the feasibility of implementing this practice in agriculture.

This research has shown that studies are needed to understand not only the behavior of insects in floristically diverse vs. depauperate landscapes, but also how anthropogenic manipulation can affect ecological interactions among crops and pollinators. In attempting to increase pollinators by promoting the presence of weeds, it is inevitable that interactions between weeds and other species will occur, and should be investigated in various crops, as well as monitored in adjacent natural systems. The best management practices moving forward are to quantify the economic ramifications accompanying the increased habitat complexity provided by weeds, and in cases where there are benefits to this approach, to take advantage of the free services they may provide.

## Figures and Tables

**Figure 1 insects-12-01114-f001:**
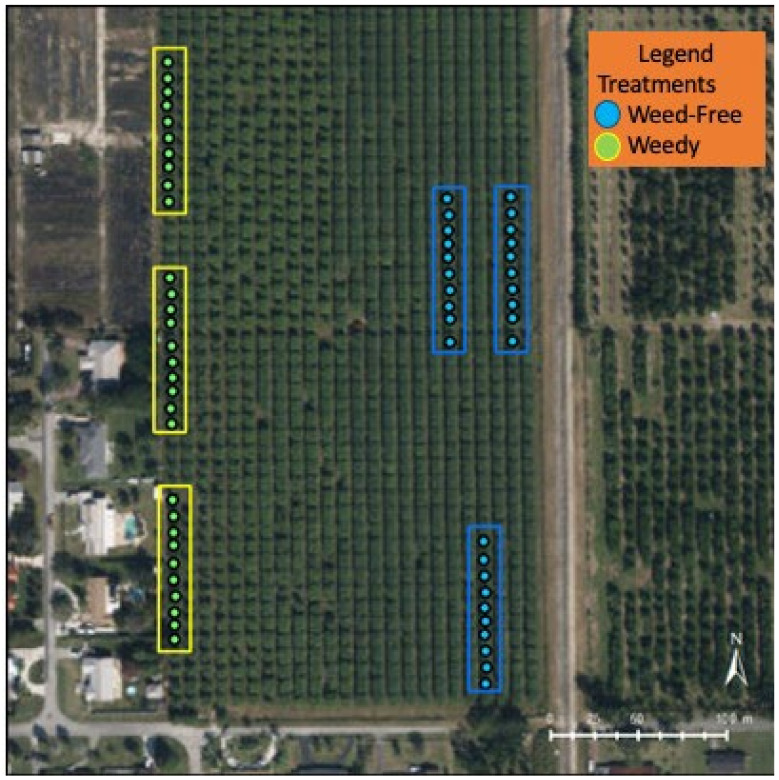
Map of weedy and weed-free trees on the farm in Homestead, FL, USA.

**Figure 2 insects-12-01114-f002:**
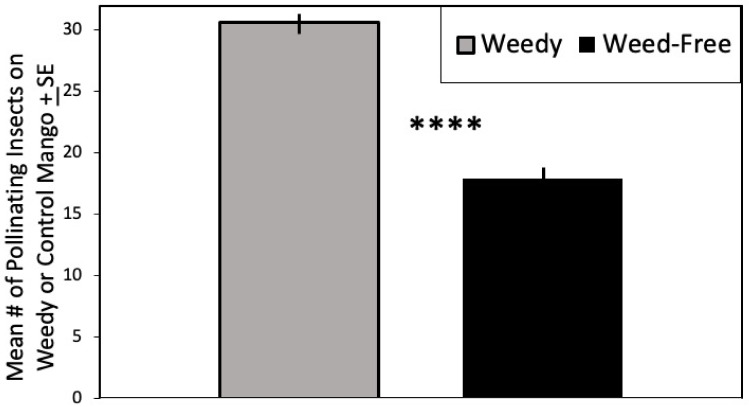
Mean Number of pollinating insects on mango by treatment ± SE. Significant difference indicated above each table. **** *p* ≤ 0.0001.

**Figure 3 insects-12-01114-f003:**
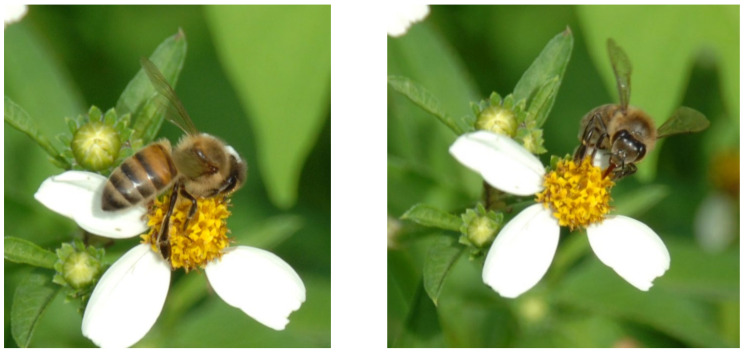
Honeybees (*Apis mellifera*) on *Bidens alba*, a common native weed in Florida.

**Figure 4 insects-12-01114-f004:**
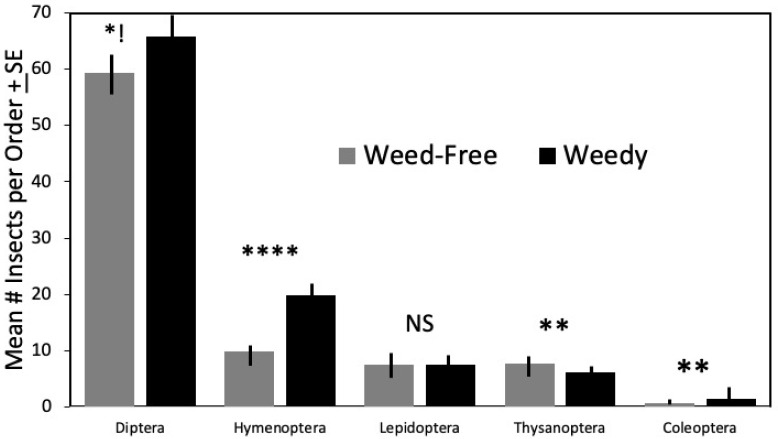
Mean number of insects observed in each order of potential pollinators on mango by treatment ± SE. Significant difference indicated above each type: NS *p* > 0.05, ** *p* ≤ 0.01, **** *p* ≤ 0.0001. Diptera *!: *p* = 0.057.

**Figure 5 insects-12-01114-f005:**
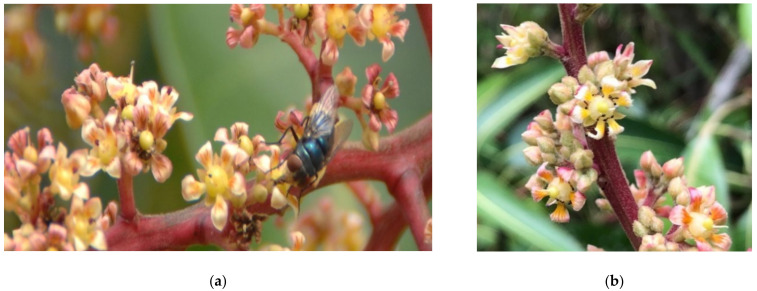
Flower visitors on mango blossoms. (**a**) Blow Fly (Calliphoridae). (**b**) Ants (Formicidae).

**Figure 6 insects-12-01114-f006:**
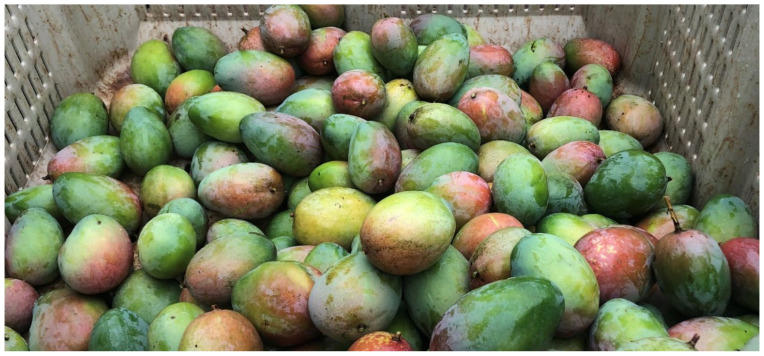
Mango “Keitt” harvest.

**Figure 7 insects-12-01114-f007:**
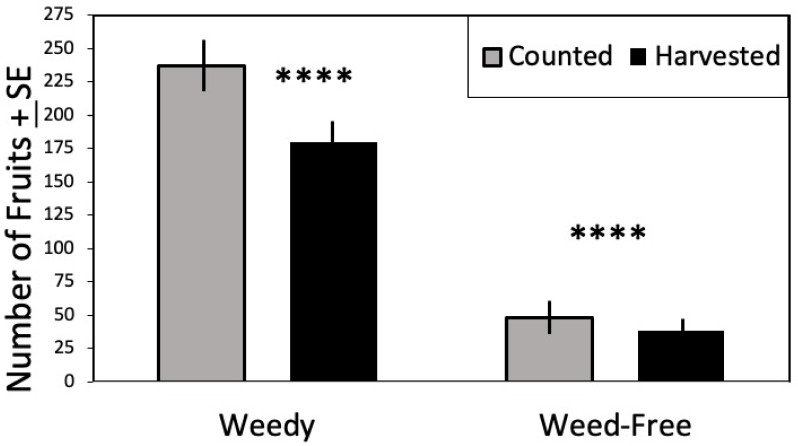
Mean number of mango fruit harvested or counted by treatment ± SE. Significant difference indicated above each Treatment **** *p* ≤ 0.0001.

**Table 1 insects-12-01114-t001:** Pollinators and fruit yield on mango trees growing with or without weeds. Significant differences as shown by GLM Multivariate ANOVA (MANOVA) have *p* values in bold.

Type	Mean	Std Deviation	N	Std Error	F	Df	*p* Value
Pollinators					31.109	1, 57	<0.0001
Weedy	30.47	1.93	30	0.35			
Weed-Free	18.10	3.24	30	0.59			
Feeding in Flower					45.934	1, 56	<0.0001
Weedy	25.90	8.93	30	1.62			
Weed-Free	10.62	8.46	29	1.57			
Harvested Fruit					181.317	1, 57	<0.0001
Weedy	179.37	65.34	30	11.88			
Weed-Free	37.67	14.63	30	2.66			
Counted Fruit					89.344	1, 57	<0.0001
Weedy	236.14	99.84	30	18.15			
Weed-Free	47.52	37.96	30	6.90			

**Table 2 insects-12-01114-t002:** Insect orders on mango with or without weeds. Significant differences as shown by GLM Multivariate ANOVA (MANOVA) have *p* values in bold.

ORDER	Mean	Std Dev	N	Std Error	F	Df	*p*
DIPTERA					3.792	1, 56	0.057
Weedy	65.77	10.59	30	1.93			
Weed-Free	59.34	14.94	29	2.77			
HYMENOPTERA					80.07	1, 56	<0.0001
Weedy	19.87	5.19	30	0.94			
Weed-Free	9.79	3.39	29	0.63			
LEPIDOPTERA					0.001	1, 56	0.970
Weedy	7.50	2.64	30	0.48			
Weed-Free	7.59	4.40	29	0.82			
COLEOPTERA					10.588	1, 56	0.002
Weedy	2.47	1.31	30	0.24			
Weed-Free	0.55	0.83	29	0.15			
THYSANOPTERA					8.208	1, 56	0.006
Weedy	6.10	2.19	30	0.40			
Weed-Free	7.66	2.24	29	0.42			

**Table 3 insects-12-01114-t003:** Insect families on mango that show significant differences in numbers on mango trees growing with/without weeds in Homestead, FL, USA.

ON MANGO:	Mean	Std Deviation	N	Std Error Mean	t	Df	*p* Value
Apidae					4.08	55	0.000
Weedy	4.17	2.12	30	0.39			
Weed-free	2.04	1.79	27	0.34			
Calliphoridae					2.03	53	0.048
Weedy	4.07	2.14	28	0.41			
Weed-free	3.00	1.75	27	0.34			
Chalcididae					2.61	29	0.003
Weedy	2.10	1.12	20	0.25			
Weed-free	1.18	0.41	11	0.12			
Chironomidae					−2.5	58	0.016
Weedy	4.53	1.65	30	0.30			
Weed-free	6.00	2.75	30	0.50			
Drosophilidae					−2.68	33	0.011
Weedy	1.28	0.58	18	0.14			
Weed-free	2.06	1.09	17	0.26			
Lycaenidae					2.18	48	0.031
Weedy	2.42	1.17	26	0.23			
Weed-free	1.83	0.64	24	0.13			
Muscidae					3.50	57	0.001
Weedy	8.27	3.81	30	0.70			
Weed-free	5.10	3.09	29	0.57			
Syrphidae					3.34	44	0.002
Weedy	6.78	2.93	27	0.56			
Weed-free	4.57	1.91	30	0.35			
Vespidae					2.2	18	0.041
Weedy	1.94	1.39	16	0.35			
Weed-free	1.13	0.35	8	0.13			

**Table 4 insects-12-01114-t004:** Weed family and species in mango farm in Homestead, FL, USA.

Family	Species	Common Name	Native/Invasive
Anacardiaceae	*Ruellia blechum*	Green Shrimp Plant	Non-native Cat II
	*Ruellia ciliatiflora*	Hairy Flower Wild Petunia	Non-native
	*Schinus terebinthifolia*	Brazilian Pepper tree	Non-native Cat I
Arecaceae	Not determined	Various indet. palms	Native & non-native
Asteraceae	*Bidens alba*	Spanish Needles	Native
	*Conoclinium coelestinum*	Blue Mistflower	Native
	*Emilia fosbergii*	Florida Tasselflower	Non-native
	*Parthenium hysterophorus*	Santa Maria Feverfew	Non-native
	*Conyza canadensis*	Canadian Horseweed	Native
	*Ageratum houstonianum*	Floss Flower	Non-native
Brassicaceae	*Lepidium virginicum*	Virginia Pepperweed	Native
	*Lepidium densiflorum*	Common Pepperweed	Non-native
Burseraceae	*Bursera simaruba*	Gumbo Limbo	Native
Commelinaceae	*Commelina diffusa*	Climbing Day flower	Non-native
Convolvulaceae	*Ipomoea hederifolia*	Scarlet Morning Glory	Native
	*Ipomoea indica*	Blue Morning Glory	Native
Cucurbitaceae	*Melothria pendula*	Creeping Cucumber	Native
Cyperaceae	*Cyperus croceus*	Baldwin’s Flatsedge	Native
Euphorbiaceae	*Acalypha ostryifolia*	Hophornbeam Copperleaf	Native
	*Acalypha setosa*	Copperleaf	Non-native
	*Euphorbia heterophylla*	Mexican Fireplant	Native
	*Euphorbia hyssopifolia*	Hyssop Spurge	Native
	*Acalypha arvensis*	Field Copperleaf	Non-native
	*Poinsettia cyathophora*	Wild Poinsettia	Native
	*Euphorbia hirta*	Asthma Plant	Native
Fabaceae	*Rhynchosia minima*	Least Snout-Bean	Native
	*Crotalaria incana*	Shake-shake	Non-native
	*Leucaena leucocephala*	White Lead tree	Non-native
	*Desmanthus virgatus*	Wild Tantan	Non-native
	*Desmodium incanum*	Creeping Beggarweed	Non-native
	*Macroptilium lathyroides*	Phasey Bean	Non-native
	*Indigofera spicata*	Creeping Indigo	Non-native
Geraniaceae	*Geranium carolinianum*	Carolina Geranium	Native
Lamiaceae	*Salvia occidentalis*	West Indian Sage	Native
Malvaceae	*Sida ulmifolia*	Common Wire Weed	Native
Moraceae	*Ficus aurea*	Strangler Fig	Native
Oleaceae	*Jasminum dichotomum*	Gold Coast Jasmine	Non-native Cat I
	*Jasminum fluminense*	Brazilian Jasmine	Non-native Cat I
Oxalidaceae	*Oxalis debilis*	Wood Sorrel	Non-native
Papaveraceae	*Argemone mexicana*	Mexican Prickly Poppy	Native
	*Fumaria officinalis*	Common Fumitory	Non-native
Phyllanthaceae	*Phyllanthus amarus*	Carry-me Seed	Non-native
Poaceae	*Paspalum conjugatum*	Hilo grass	Native
	*Urochloa maxima*	Guinea grass	Non-native Cat II
	*Sporobolus jacquemontii*	American Rat’s Tail Grass	Non-native Cat I
	*Digitaria ciliaris*	Southern Crabgrass	Native
	*Melinis repens*	Natal grass	Non-native Cat I
	*Stenotaphrum secundatum*	St. Augustine grass	Native
	*Sorghum halepense*	Johnson grass	Non-native
	*Panicum maximun*	Guinea grass	Non-native
	*Cenchrus* sp.	Sandbur	Native
	*Neyraudia reynaudiana*	Burma Reed	Non-native Cat I
Psilotaceae	*Psilotum nudum*	Whisk Fern	Native
Rubiaceae	*Spermacoce remota*	Woodland False Buttonweed	Native
	*Spermacoce verticillata*	shrubby false buttonweed	Non-native Cat II
	*Richardia brasiliensis*	Brazil Pusley	Non-native
	*Richardia scabra*	Florida Pusley	Non-native
	*Richardia grandiflora*	Large-flower Pusley	Non-native Cat II
Sapindaceae	*Cardiospermum corindum*	Soapberry	Native
Solanaceae	*Solanum americanum*	American Black Nightshade	Native
Urticaceae	*Pouzolzia zeylanica*	Pouzolzia	Non-native
	*Laportea aestuans*	West Indian Wood-nettle	Non-native
	*Pilea microphylla*	Artillery plant	Native
Verbenaceae	*Lantana camara*	Common Lantana	Non-native
Vitaceae	*Cissus verticillata*	Possum Grape Vine	Native
	*Ampelopsis cordata*	Heartleaf Peppervine	Native
	*Parthenocissus quinquefolia*	Virginia Creeper	Non-native
Zamiaceae	*Zamia furfuracea*	Cardboard Palm	Non-native

## Data Availability

Data to be deposited in the FIU Research Data Portal, a publicly accessible repository: https://doi.org/10.34703/gzx1-9v95/SWGODJ.

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
