# Peer review of "Weeds Enhance Pollinator Diversity and Fruit Yield in Mango"

_insects, 2021, doi:10.3390/insects12121114_

Round 1

Reviewer 1 Report

The manuscript “Weeds enhance pollinators diversity and fruit yield in mango” presents the results of a small field experiment showing that weeds presence increases insect (pollinator) presence, leading to better fruit yield. I see no major flaw with the study itself, it is a small-scale study (e.g. little replication, only one field site) but still deserves to be published and adds data to an existing body of evidence.

The text needs to be heavily restructured and shortened. There is a lot of repetitions, and the manuscript does not follow a classic article structure. For instance, results are discussed in the result section - it makes it difficult to efficiently extract information from the manuscript.

Introduction

I suggest that the authors restructure and shorten the introduction. At the moment, it has no obvious structure and contains a lot of repetitions. The authors could for instance articulate their introduction as following: 1) Importance of pollination for crops, 2) Pollinators are declining, 3) Weeds increase floral resources and pollinator density, 4) Review on work that has been done on mango pollination, 5) aim of the study, hypothesis. Restructuring the introduction in a more logical way will help the authors to get rid of repetitions.

Methods

On Figure 1, the western side of the plot looks different from the eastern side (i.e. more sparse trees). Is there any explanation for that, given that the experimental trees are not randomised?

Results

The results are discussed in the result section, which is unusual. Sections lines 297-300, 309-328, 336-347, 361-368, 374-380, 396-404 should be moved to the discussion.

Discussion

No major comment, apart from condensing/restructuration with the current contents of the result sections.

  • Lines 444-446: There is no data to back that, state results in the results section or make it clear it is just an observation

Minor comments

  • Line 21: "compare the insects" needs more detail. Is it about diversity or quantity?
  • Line 67: "the expansion of monoculture practices" should have a reference too
  • Line 115: "on flowers" should come before "for feeding"
  • Line 295: "flies provided many flower visitors" is a bit awkward

Author Response

please see our responses in the attached document

Reviewer 2 Report

Review of manuscript nsects-1466937 “Weeds enhance Pollinator Diversity and Fruit Yield in Mango.

Overall, there is good data and evidence in the manuscript for the support of conclusion. My main concern is with the lack of details in the methods section as well as the lengthy introduction which at times rambles and lacks focus. Finally, the authors do not address current relevant hypothesis.

The introduction should be shortened and focus on the specific system being addressed.

Details about how weed abundances were estimated must be included. Is there a given abundance or diversity of weeds that must be reached in order to be considered weedy? How much variation of weed was observed in the treatment group?

How did you deal with issues of spatial dependency in the study?

How were insects collected? How many? How were those preserved? Are there voucher specimens, and if so, where were they deposited?

Finally, the ecological intensification hypothesis is extremely relevant to this work. I strongly encourage the authors to read up on it and to consider how their work can contribute to the overall discussion.

Ecological Intensification general:

Doré, T., Makowski, D., Malézieux, E., Munier-Jolain, N., Tchamitchian, M. and Tittonell, P., 2011. Facing up to the paradigm of ecological intensification in agronomy: revisiting methods, concepts and knowledge. European Journal of Agronomy, 34(4), pp.197-210.

Kleijn, D., Bommarco, R., Fijen, T.P., Garibaldi, L.A., Potts, S.G. and Van Der Putten, W.H., 2019. Ecological intensification: bridging the gap between science and practice. Trends in ecology & evolution, 34(2), pp.154-166.

Ecological Intensification pollination:

Kovács‐Hostyánszki, A., Espíndola, A., Vanbergen, A.J., Settele, J., Kremen, C. and Dicks, L.V., 2017. Ecological intensification to mitigate impacts of conventional intensive land use on pollinators and pollination. Ecology Letters, 20(5), pp.673-689.

Albrecht, M., Kleijn, D., Williams, N.M., Tschumi, M., Blaauw, B.R., Bommarco, R., Campbell, A.J., Dainese, M., Drummond, F.A., Entling, M.H. and Ganser, D., 2020. The effectiveness of flower strips and hedgerows on pest control, pollination services and crop yield: a quantitative synthesis. Ecology letters, 23(10), pp.1488-1498.

Author Response

please see our replies in the attached document. Thank you!

Round 2

Reviewer 1 Report

The authors seem to have taken my suggestions into account, no further comment.

Reviewer 2 Report

The authors have included all the required changes.  The manuscript is significantly improved and should make a good addition to our understanding of ecological supplementation.